# Predictive value of pulse oximetry for mortality in infants and children presenting to primary care with clinical pneumonia in rural Malawi: A data linkage study

Tim Colbourn[1]*, Carina King[1,2], James Beard[1], Tambosi Phiri[3], Malizani Mdala[3], Beatiwel Zadutsa[3], Charles Makwenda[3], Anthony Costello[1], Norman Lufesi[4], Charles Mwansambo[4], Bejoy Nambiar[5], Shubhada Hooli[6], Neil French[7], Naor Bar Zeev[7,8,9], Shamim Ahmad Qazi[10¤], Yasir Bin Nisar[11], Eric D. McCollum[9,12]

1 Institute for Global Health, University College London, London, United Kingdom, 2 Department of Global Public Health, Karolinska Institutet, Stockholm, Sweden, 3 Parent and Child Health Initiative, Lilongwe, Malawi, 4 Ministry of Health, Lilongwe, Malawi, 5 UNICEF, Lilongwe, Malawi, 6 Department of Pediatrics, Section of Emergency Medicine, Baylor College of Medicine, Houston, Texas, United States of America, 7 Institute of Infection & Global Health, University of Liverpool, Liverpool, United Kingdom, 8 Malawi-Liverpool-Wellcome Trust Clinical Research Programme, Blantyre, Malawi, 9 Department of International Health, Johns Hopkins Bloomberg School of Public Health, Baltimore, Maryland, United States of America, 10 Department of Maternal, Newborn, Child and Adolescent Health, World Health Organization, Geneva, Switzerland, 11 Department of Maternal, Newborn, Child and Adolescent Health and Ageing, World Health Organization, Geneva, Switzerland, 12 Global Program in Pediatric Respiratory Sciences, Eudowood Division of Pediatric Respiratory Sciences, Department of Pediatrics, Johns Hopkins School of Medicine, Baltimore, Maryland, United States of America

¤ Current address: Retired, Geneva, Switzerland.
* t.colbourn@ucl.ac.uk

## Abstract

### Background

The mortality impact of pulse oximetry use during infant and childhood pneumonia management at the primary healthcare level in low-income countries is unknown. We sought to determine mortality outcomes of infants and children diagnosed and referred using clinical guidelines with or without pulse oximetry in Malawi.

### Methods and findings

We conducted a data linkage study of prospective health facility and community case and mortality data. We matched prospectively collected community health worker (CHW) and health centre (HC) outpatient data to prospectively collected hospital and community-based mortality surveillance outcome data, including episodes followed up to and deaths within 30 days of pneumonia diagnosis amongst children 0–59 months old. All data were collected in Lilongwe and Mchinji districts, Malawi, from January 2012 to June 2014. We determined differences in mortality rates using <90% and <93% oxygen saturation ($SpO_2$) thresholds and World Health Organization (WHO) and Malawi clinical guidelines for referral. We used unadjusted and adjusted (for age, sex, respiratory rate, and, in analyses of HC data only, Weight for Age Z-score [WAZ]) regression to account for interaction between $SpO_2$ threshold (pulse

**Data Availability Statement:** We are unable to make the original data available as it contains personally identifiable information, which is key to

the matching process. Our data linkage study uses data from three publications (references [6], [13], and [14] of the paper): 6. McCollum ED, King C, Deula R, Zadutsa B, Mankhambo L, Nambiar B, et al. Outpatient pulse oximetry implementation with rural facility and community health workers during three years of child pneumonia care in two central Malawi districts. Bulletin of the World Health Organisation. 2016;94:893-902. 13. McCollum ED, Nambiar B, Deula R, Zadutsa B, Bondo A, King C, et al. Impact of the 13-valent Pneumococcal Conjugate Vaccine on Clinical and Hypoxemic Childhood Pneumonia over Three Years in Central Malawi: An observational study. PLoS One. 2017;DOI:10.1371/journal.pone.0168209 January 4, 2017. 14. Bar-Zeev N, King C, Phiri T, Beard J, Mvula H, Crampin AC, et al. Impact of monovalent rotavirus vaccine on diarrhoea-associated post-neonatal infant mortality in rural communities in Malawi: a population-based birth cohort study. The Lancet Global health.

**Funding:** A grant from the Bill & Melinda Gates Foundation (BMGF) https://www.gatesfoundation.org/ (OPP1106190) to the World Health Organization (YBN and SAQ) funded the data linkage of the morbidity surveillance originally funded by a BMGF grant (#23591) to AC and the mortality surveillance funded by a Wellcome Trust https://wellcome.ac.uk/ Programme Grant (WT091909/B/10/Z0) to NF. The funders had no role in study design, data collection and analysis, decision to publish, or preparation of the manuscript. The corresponding author TC and authors CK, JB, and EDM had full access to all the data and had final responsibility to submit for publication.

**Competing interests:** The authors have declared that no competing interests exist.

**Abbreviations:** CHW, community health worker; DOR, diagnostic odds ratio; GLM, generalised linear model; HC, health centre; iCCM, integrated community case management; IMCI, integrated management of childhood illness; LMICs, low-income and middle-income countries; RR, Risk Ratio; SpO2, oxygen saturation; STROBE, Strengthening the Reporting of Observational Studies in Epidemiology; WAZ, Weight for Age Z-score; WHO, World Health Organization.

oximetry) and clinical guidelines, clustering by child, and CHW or HC catchment area. We matched CHW and HC outpatient data to hospital inpatient records to explore roles of pulse oximetry and clinical guidelines on hospital attendance after referral. From 7,358 CHW and 6,546 HC pneumonia episodes, we linked 417 CHW and 695 HC pneumonia episodes to 30-day mortality outcomes: 16 (3.8%) CHW and 13 (1.9%) HC patients died. $SpO_2$ thresholds of <90% and <93% identified 1 (6%) of the 16 CHW deaths that were unidentified by integrated community case management (iCCM) WHO referral protocol and 3 (23%) and 4 (31%) of the 13 HC deaths, respectively, that were unidentified by the integrated management of childhood illness (IMCI) WHO protocol. Malawi IMCI referral protocol, which differs from WHO protocol at the HC level and includes chest indrawing, identified all but one of these deaths. $SpO_2 < 90\%$ predicted death independently of WHO danger signs compared with $SpO_2 \geq 90\%$: HC Risk Ratio (RR), 9.37 (95% CI: 2.17–40.4, p = 0.003); CHW RR, 6.85 (1.15–40.9, p = 0.035). $SpO_2 < 93\%$ was also predictive versus $SpO_2 \geq 93\%$ at HC level: RR, 6.68 (1.52–29.4, p = 0.012). Hospital referrals and outpatient episodes with referral decision indications were associated with mortality. A substantial proportion of those referred were not found admitted in the inpatients within 7 days of referral advice. All 12 deaths in 73 hospitalised children occurred within 24 hours of arrival in the hospital, which highlights delay in appropriate care seeking. The main limitation of our study was our ability to only match 6% of CHW episodes and 11% of HC episodes to mortality outcome data.

## Conclusions

Pulse oximetry identified fatal pneumonia episodes at HCs in Malawi that would otherwise have been missed by WHO referral guidelines alone. Our findings suggest that pulse oximetry could be beneficial in supplementing clinical signs to identify children with pneumonia at high risk of mortality in the outpatient setting in health centres for referral to a hospital for appropriate management.

## Author summary

### Why was this study done?

- Pneumonia is a leading cause of death of children under 5 years old, and early identification and treatment of severe cases is required to prevent deaths.

- Pulse oximetry is more sensitive at detecting hypoxaemia than clinical signs alone and therefore can potentially prevent more deaths from pneumonia.

- There is a lack of evidence of the effect on child deaths of pulse oximetry use by healthcare workers in informal community settings and at formal primary care clinics, and this study sought to fill this evidence gap.

## What did the researchers do and find?

- We linked Malawian community health worker and health centre outpatient data to hospital and community mortality data to determine the mortality outcomes for children with pneumonia identified by pulse oximetry or clinical signs or both as outpatients.

- We show that pulse oximetry identified fatal episodes of childhood pneumonia that did not have identified clinical signs.

- Pulse oximetry readings of less than 90% oxygen saturation ($SpO_2$) identified 6% of deaths at community health worker level (1/16) and 23% of deaths at health centre level (3/13) not identified by clinical signs.

- Increasing the threshold to less than 93% $SpO_2$, pulse oximetry identified 1 additional death (1/13, 7.7% of deaths) not identified by clinical signs at the health centre level only.

- All of the health centre deaths identified by pulse oximetry except one were also identified by chest indrawing in this high-mortality setting.

## What do these findings mean?

- Our findings suggest that pulse oximetry could be beneficial in supplementing clinical signs to identify children with pneumonia at high risk of mortality in the outpatient setting in health centres for referral to a hospital for appropriate management.

- In high-mortality settings in low- and middle-income countries, in the absence of pulse oximetry, presence of chest indrawing could potentially be explored as a referral sign to a hospital but needs further research in routine settings.

## Introduction

Pneumonia remains a leading cause of death in children under 5, especially in low-income and middle-income countries (LMICs), with around 800,000 pneumonia-related deaths a year globally [1]. Incidence of clinical pneumonia is estimated to be as high as 500 episodes per 1,000 child-years in some regions, with an average of 122 episodes per 1,000 child-years in Africa in 2017 [2]. Although hospitalisation rates are increasing and hospital case fatality rates are decreasing, case fatality rates are still typically around 3%–5% in LMICs [3]. Many deaths still occur at home after care seeking [4].

Early identification and action is required to prevent more pneumonia-related deaths that currently occur in hospital, often because of late presentation [5], and at home, often because their illness severity is unrecognised by primary care healthcare providers or there are barriers to accessing secondary care.

As an objective measurement of physiological illness severity, noninvasive peripheral oxygen saturation ($SpO_2$) measurement by pulse oximeters at outpatient primary care and first level health facilities has a potential role to aid early recognition and referral of severe

pneumonia episodes for oxygen and injectable antibiotic treatment [6, 7]. The current 2014 World Health Organization (WHO) Integrated Management of Childhood Illness (IMCI) chart booklet includes pulse oximetry as optional rather than mandatory and stipulates a threshold of <90% to indicate hypoxaemia requiring immediate referral to hospital [8]. However, $SpO_2 < 93\%$ has also been shown to predict fatality in high-mortality settings [9, 10].

Evidence linking outpatient primary care $SpO_2$ measurement to hospital referral and fatality is lacking [11, 12]. We aim to address this gap by linking data from previously described community and hospital based morbidity and mortality surveillance studies [9, 13–16] with a concurrent study in the same district involving outpatient pulse oximeter use at the community health worker (CHW) and health centre (HC) levels [6]. We assess the potential added value of outpatient pulse oximetry to hospital referrals and mortality outcomes using the current <90% $SpO_2$ threshold and a <93% threshold, in conjunction with pre-2018 Malawi guidelines in use at the time of data collection (hereafter 'Malawi guidelines') [17]. Additionally, we explored the theory that the inability to obtain a pulse oximetry measurement may be associated with mortality [18]. Malawi guidelines for HC cases, unlike the WHO IMCI 2014 chart booklet, included mandatory hospital referral for 2- to 59-month–old children with chest indrawing (i.e., bilateral inward pulling of the lower anterior subcostal tissue during inspiration). For CHW cases, both WHO integrated community case management (iCCM) [19] and Malawi guidelines [17] indicate referral of infants and children with chest indrawing or danger signs (Table 1). This study was agreed on after an exploratory meeting by WHO on IMCI danger signs for pneumonia [12].

We hypothesised that pulse oximetry use for outpatients could identify children at risk of dying who would not be identified by clinical signs alone.

## Methods

Our objectives were to determine whether pulse oximetry at outpatient CHW and HC primary care levels identifies infant and child pneumonia patients for referral to hospital independently of clinical signs included in the Malawi and WHO guidelines for CHW and HC patients (Table 1) and the fatality outcomes at 30 days postdiagnosis. To determine fatality, we linked CHW and HC outpatient pneumonia data sets of 0- to 59-month–olds in Lilongwe and Mchinji districts, Malawi (Fig 1) from 1st Jan 2012 to 30th June 2014 [6, 13] to hospital [13] data for the same time period. We also linked the outpatient data to community-surveillance mortality data available for the Mchinji district only [14]. We developed and then followed our prespecified analysis plan (S1 Appendix) as far as we were able given the limitations of the final matched data set. We added the sensitivity and specificity analyses at the request of the statistical reviewer. This study is reported as per the Strengthening the Reporting of Observational Studies in Epidemiology (STROBE) guideline (S1 Checklist).

### Setting

In the Malawi healthcare system, children are intended to access care at either village clinics or HCs. At the village clinic level, if the child is found to be referral eligible, then the child is expected to be referred by the CHW to either the HC or hospital. At the HC level, children meeting referral criteria are referred to hospital.

### Sample characteristics

Children were aged 0–59 months with a clinical pneumonia diagnosis according to routine data prospectively collected by 38 CHWs and providers from 18 HCs in rural Lilongwe and Mchinji districts, Malawi (Table 1) [6].

**Table 1. Matching of outpatient child pneumonia episodes to mortality outcome data.**

| | | CHW Episodes with Outcome Data (N = 417), n (%) | CHW Episodes without Outcome Data (N = 6,941), n (%) | p-Value[a] | HC Episodes with Outcome Data (N = 695), n (%) | HC Episodes without Outcome Data (N = 5,761), n (%) | p-Value[a] |
|---|---|---|---|---|---|---|---|
| Outcome: | Death[i] | 16 (3.8%) | no data | | 13 (1.9%) | no data | |
| | Survival | 401 (96.2%) | no data | | 682 (98.1%) | no data | |
| SpO$_2$ | <90% | 7 (1.9%) | 79 (1.2%) | 0.240 | 65 (10.1%) | 543 (10.2%) | 0.971 |
| | ≥90% | 362 (98.1%) | 6,496 (98.8%) | | 578 (89.9%) | 4,804 (89.8%) | |
| | <93% | 14 (3.8%) | 520 (7.9%) | 0.004 | 128 (19.9%) | 1,056 (19.8%) | 0.925 |
| | ≥93% | 355 (96.2%) | 6,055 (92.1%) | | 515 (80.1%) | 4,291 (80.2%) | |
| | Failed measurement | 48 (11.5%) | 366 (5.3%) | <0.001 | 52 (7.5%) | 414 (7.2%) | 0.776 |
| | Measured | 369 (88.5%) | 6,575 (94.7%) | | 643 (92.5%) | 5,347 (92.8%) | |
| Chest indrawing | | 16 (3.8%) | 105 (1.5%) | <0.001 | 241 (34.7%) | 1,457 (25.3%) | <0.001 |
| Danger signs[ii] (WHO guidelines clinically eligible for referral = same as Malawi guidelines at CHW level) | | 41 (9.8%) | 962 (13.9%) | 0.020 | 110 (15.8%) | 612 (10.6%) | <0.001 |
| Abnormally sleepy | | 2 (0.5%) | 11 (0.2%) | 0.129 | 6 (0.9%) | 74 (1.3%) | 0.343 |
| Baby apnoeic | | no data | no data | | 9 (1.3%) | 18 (0.3%) | <0.001 |
| Had convulsions | | 1 (0.2%) | 266 (3.8%) | <0.001 | 5 (0.7%) | 58 (1.0%) | 0.467 |
| Not breastfeeding or drinking | | 3 (0.7%) | 73 (1.1%) | 0.514 | 20 (2.9%) | 143 (2.5%) | 0.530 |
| Vomiting everything | | 17 (4.1%) | 554 (8.0%) | 0.004 | no data | no data | |
| Stridor when calm | | no data | no data | | 17 (2.5%) | 165 (2.9%) | 0.529 |
| HIV exposure/infection | | no data | no data | | 20 (3.0%) | 93 (1.7%) | 0.018 |
| Swelling of both feet | | 6 (1.4%) | 77 (1.1%) | 0.536 | no data | no data | |
| Malnutrition (clinical) | | no data | no data | | 7 (1.0%) | 40 (0.7%) | 0.359 |
| SAM (MUAC < 11.5 cm, ≥6 months old) | | 6 (2.1%) | 31 (0.5%) | 0.001 | 10 (2.1%) | 49 (1.1%) | 0.063 |
| Malawi guidelines clinically eligible for referral[iii] | | 41 (9.8%) | 962 (13.9%) | 0.020 | 275 (39.6%) | 1,690 (29.3%) | <0.001 |
| Sex | Male (% not missing) | 218 (53.2%) | 3,325 (48.4%) | 0.062 | 363 (56.2%) | 2,751 (51.6%) | 0.027 |
| | Female (% not missing) | 192 (46.8%) | 3,541 (51.6%) | | 283 (43.8%) | 2,581 (48.4%) | |
| | Missing data (% total) | 7 (1.7%) | 74 (1.1%) | | 49 (7.1%) | 429 (7.5%) | |
| Age [months] mean (SD, min−max) | | 10 (7, 2−58) | 24 (15, 0−59) | <0.001 | 8 (7, 1−48) | 16 (12, 0−59) | <0.001 |
| Missing data (n, %) | | 16 (3.8%) | 682 (9.8%) | | 0 (0%) | 0 (0%) | |
| Respiratory Rate[iv]: | Normal (% not missing) | 3 (0.8%) | 84 (1.4%) | 0.380 | 27 (4.1%) | 164 (3.0%) | 0.125 |
| | Fast (% not missing) | 359 (93.0%) | 5,716 (93.6%) | | 571 (85.7%) | 4,665 (84.9%) | |
| | Very fast (% not missing) | 24 (6.2%) | 309 (5.1%) | | 68 (10.2%) | 669 (12.2%) | |
| | Missing data (% total) | 31 (7.4%) | 832 (12.0%) | | 29 (4.2%) | 262 (4.6%) | |

(*Continued*)

**Table 1.** (Continued)

| | | CHW Episodes with Outcome Data (N = 417), n (%) | CHW Episodes without Outcome Data (N = 6,941), n (%) | p-Value[a] | HC Episodes with Outcome Data (N = 695), n (%) | HC Episodes without Outcome Data (N = 5,761), n (%) | p-Value[a] |
|---|---|---|---|---|---|---|---|
| WAZ[v]: | Normal (>−2 z-scores) | no data | no data | | 561 (88.6%) | 4,395 (84.3%) | 0.011 |
| | Low (−3 to −2 z-scores) | | | | 44 (7.0%) | 545 (10.5%) | |
| | Severely low (<−3 z-scores) | | | | 28 (4.4%) | 276 (5.3%) | |
| | Missing data (n, %) | | | | 62 (8.9%) | 544 (9.4%) | |

[i]Within 30 days of being seen at CHW or HC level (all were 0–7 days).

[ii]A composite indicator variable coded as yes (1) if any of the danger signs in the 10 rows below are present (6 for CHWs and 8 for HCs)—this is equivalent to WHO 2014 iCCM guidelines clinically eligible for referral for community (CHW episodes) and to IMCI guidelines clinically eligible for referral for HC episodes. The 4 danger signs with no data for the CHW episodes were not assessed by the CHWs because they are not part of the iCCM guidelines: baby apnoeic and stridor because they require clinical training beyond CHW level to assess, HIV because testing is not available at community level, and 'malnutrition (clinical)' is covered by 'swelling of both feet' above. The 2 danger signs with no data for the HCs were not assessed by the HC workers because they are not part of the IMCI guidelines: 'vomiting everything' because it is covered under a full assessment of 'not breastfeeding and drinking' above and 'swelling of both feet' because it is covered by full assessment of 'malnutrition (clinical)' below. Please note danger signs denoting referral are different for 0- to 2-month–old infants, and the danger signs variables were coded for these very young infants accordingly, both for CHW and HC episodes; in particular, it is important to note that chest indrawing in 0- to 2-month–old infants is a danger sign requiring referral in IMCI (HC episodes). In iCCM guidelines, chest indrawing is a danger sign requiring referral for all children aged 0–59 months; therefore, WHO guidelines (iCCM) are the same as Malawi guidelines for CHW episodes.

[iii]A composite variable coded as yes (1) if chest indrawing or any of the danger signs in the 10 rows above are present (6 for CHWs and 8 for HCs).

[iv]Fast breathing: $\geq 60$ and $\leq 79$, $\geq 50$ and $\leq 69$, $\geq 40$ and $\leq 59$ breaths per minute for <2, 2–11, and 12–59 months of age categories; very fast breathing: $\geq 80$, $\geq 70$, and $\geq 60$ breaths per minute for <2, 2–11, and 12–59 months of age categories.

[v]WAZ calculated from age in months and weight in kilograms, using WHO growth curves for males and females separately via the zanthro user-written add-on function in Stata. Percentages without missing data shown.

[a]Chi-squared test for categorical variables (sex) missing data category excluded, *t* test for continuous variables (age, weight).

**Abbreviations:** CHW, community health worker; HC, health centre; iCCM, integrated community case management; IMCI, integrated management of childhood illness; MUAC, mid-upper arm circumference; SAM, Severe Acute Malnutrition; SpO₂, oxygen saturation; WAZ, Weight for Age Z-score; WHO, World Health Organization.

## Pulse oximetry measurements

Healthcare providers underwent a 1-day training in pulse oximetry, medical record keeping, and the definition of pneumonia at the start of the study period; they had continued support through monthly mentorship visits. SpO₂ measurements were taken using the Lifebox device (Acare Technology, Xinzhuang, Taiwan, China), with a universal adult clip probe applied to the child's big toe if less than 2 years of age or below 10 kg. Otherwise, for older or heavier children, providers were instructed to use either the big toe or an appropriately sized finger. A paediatric probe was not available during this time period. CHW and HC workers were trained and retrained in the use of pulse oximetry by a paediatric pulmonologist (EDM) as described by McCollum and colleagues [6]. Providers were trained to record measurements that demonstrated consistent plethysmography waveforms along with a stable, nondrifting SpO₂ and age-appropriate pulse rate. Given this work was conducted within a routine clinical context, providers were not required to repeat measurements meeting these quality criteria. This study showed moderate agreement in sequentially obtained SpO₂ readings between EDM and each cadre of health workers [6].

## Matching

Matching of the CHW and HC data sets with the hospital and population mortality surveillance outcome data was done using the following parameters: child name, caregiver or parent

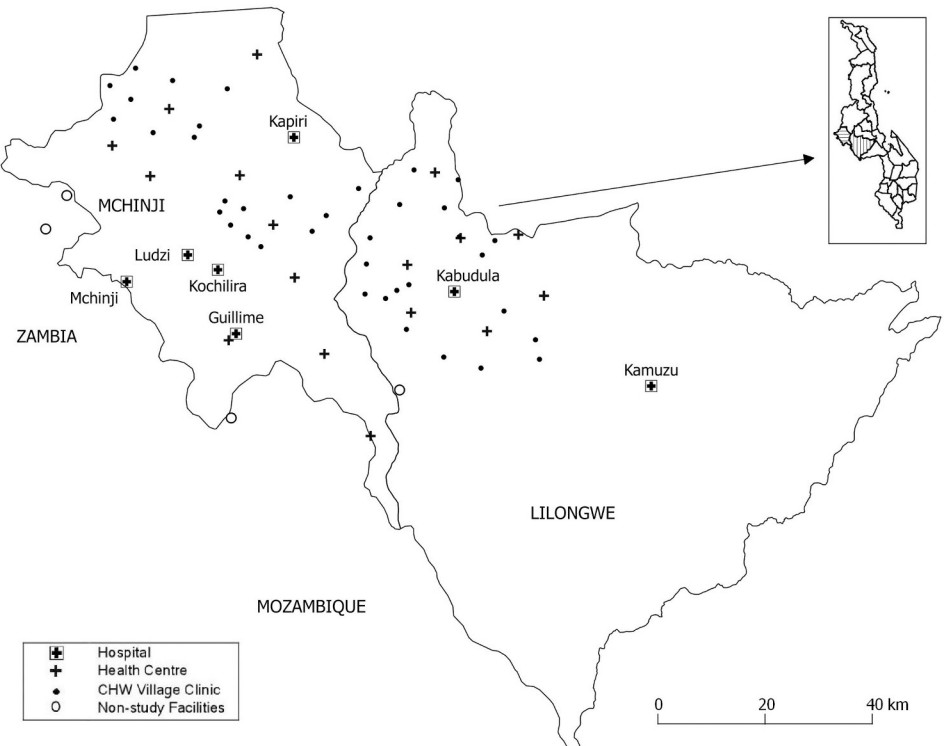

**Fig 1. Map of the study area.** CHW, community health worker.

name, age at known date or date of birth, address, and PCV13 vaccination dates, using a probabilistic algorithm (S2 Appendix, pp. 1–3). We compared outpatient pneumonia episodes successfully matched to 30-day fatality data to those remaining unmatched to assess the representativeness of the matched sample.

## Analysis

We constructed sets of 6 mutually exclusive and complete groupings of episodes according to whether they met SpO$_2$ thresholds for referral or failed attempted SpO$_2$ measurement (defined as no stable reading after 5 minutes of measurement [6]), clinical referral criteria, both, or neither (Table 2).

We described the distribution of deaths in the matched data set and crude differences in fatality for each of the 6 groupings in the SpO$_2$ and clinical guidelines exposure sets (Table 2). We determined the independent associations of SpO$_2$ and danger sign exposures on fatality, using generalised linear models (GLMs) of the binomially distributed binary outcome, with a log link (Eq 1); these are analogous to logistic regression but produce Risk Ratios (RRs), which are easier to interpret than the odds ratios produced by logistic regression [20]. We ran GLMs for the matched CHW and HC data separately, i.e., for each of the 6 exposure sets. The base-case unadjusted model using <90% SpO$_2$ and Malawi guidelines clinical referral criteria thresholds (Model M90, see Table 3), is

$$Y_1 \sim \text{binomial}(\mu_i)$$

$$\log(\mu_i) = \beta_0 + \beta_1 X_{1\_1} + \beta_2 X_{2\_1} + \beta_3 X_{1\_1} X_{2\_1} + \varepsilon, \tag{1}$$

**Table 2. Mortality outcomes by SpO₂ and danger sign exposure group sets.**

**CHW Data**

| N = 417, n (col %) | Malawi guidelines (= WHO guidelines), <90% SpO₂ threshold (this was used by the healthcare workers) | Died within 30 days, n (row %) | N = 417, n (col %) | Malawi guidelines (= WHO guidelines), <93% SpO₂ threshold | Died within 30 days, n (row %) |
|---|---|---|---|---|---|
| 329 (78.9%) | NOT Malawi clinically eligible and SpO₂ > 90% | 12 (3.6%) | 324 (77.7%) | NOT Malawi clinically eligible and SpO₂ ≥ 93% | 12 (3.7%) |
| 33 (7.9%) | Malawi clinically eligible only and SpO₂>90% | 2 (6.1%) | 31 (7.4%) | Malawi clinically eligible only and SpO₂ ≥ 93% | 1 (3.2%) |
| 4 (1.0%)[a] | SpO₂ <90% only and not Malawi clinically eligible[a] | 1 (25.0%)[a] | 9 (2.2%)[a] | SpO₂ < 93% only and not Malawi clinically eligible[a] | 1 (11.1%)[a] |
| 3 (0.7%) | SpO₂ <90% and Malawi clinically eligible | 0 (0.0%) | 5 (1.2%) | SpO₂ <93% and Malawi clinically eligible | 1 (20.0%) |
| 5 (1.2%) | failed SpO₂ measurement but Malawi clinically eligible | 0 (0.0%) | 5 (1.2%) | failed SpO₂ measurement but Malawi clinically eligible | 0 (0.0%) |
| 43 (10.3%)[b] | failed SpO₂ measurement and not Malawi clinically eligible[b] | 1 (2.3%)[b] | 43 (10.3%)[b] | failed SpO₂ measurement and not Malawi clinically eligible[b] | 1 (2.3%)[b] |

**HC Data**

| N = 695, n (col %) | Malawi guidelines, <90% SpO₂ threshold (this was used by the healthcare workers) | Died within 30 days, n (row %) | N = 695, n (col %) | Malawi guidelines, <93% SpO₂ threshold | Died within 30 days, n (row %) |
|---|---|---|---|---|---|
| 387 (55.7%) | NOT Malawi clinically eligible and SpO₂ > 90% | 2 (0.5%) | 363 (52.2%) | NOT Malawi clinically eligible and SpO₂ ≥ 93% | 2 (0.6%) |
| 191 (27.5%) | Malawi clinically eligible only and SpO₂ > 90% | 4 (2.1%) | 152 (21.9%) | Malawi clinically eligible only and SpO₂ ≥ 93% | 3 (2.0%) |
| 15 (2.2%)[a] | SpO₂ <90% only and not Malawi clinically eligible[a] | 0 (0.0%)[a] | 39 (5.6%)[a] | SpO₂ < 93% only and not Malawi clinically eligible[a] | 0 (0.0%)[a] |
| 50 (7.2%) | SpO₂ < 90% and Malawi clinically eligible | 6 (12.0%) | 89 (12.8%) | SpO₂ < 93% and Malawi clinically eligible | 7 (7.9%) |
| 34 (4.9%) | failed SpO₂ measurement but Malawi clinically eligible | 0 (0.0%) | 34 (4.9%) | failed SpO₂ measurement but Malawi clinically eligible | 0 (0.0%) |
| 18 (2.6%)[b] | failed SpO₂ measurement and not Malawi clinically eligible[b] | 1 (5.6%)[b] | 18 (2.6%)[b] | failed SpO₂ measurement and not Malawi clinically eligible[b] | 1 (5.6%)[b] |

| N = 695, n (col %) | WHO guidelines, <90% SpO₂ threshold | Died within 30 days, n (row %) | N = 695, n (col %) | WHO guidelines, <93% SpO₂ threshold | Died within 30 days, n (row %) |
|---|---|---|---|---|---|
| 512 (73.7%) | NOT WHO clinically eligible and SpO₂ ≥ 90% | 4 (0.8%) | 461 (66.3%) | NOT WHO clinically eligible and SpO₂ ≥ 93% | 3 (0.7%) |
| 66 (9.5%) | WHO clinically eligible only and SpO₂ ≥ 90% | 2 (3.0%) | 54 (7.8%) | WHO clinically eligible only and SpO₂ ≥ 93% | 2 (3.7%) |
| 41 (5.9%)[a] | SpO₂ < 90% only and not WHO clinically eligible[a] | 3 (7.3%)[a] | 92 (13.2%)[a] | SpO₂ < 93% only and not WHO clinically eligible[a] | 4 (4.3%)[a] |
| 24 (3.5%) | SpO₂ < 90% and WHO clinically eligible | 3 (12.5%) | 36 (5.2%) | SpO₂ < 93% and WHO clinically eligible | 3 (8.3%) |
| 20 (2.9%) | failed SpO₂ measurement but WHO clinically eligible | 0 (0.0%) | 20 (2.9%) | failed SpO₂ measurement but WHO clinically eligible | 0 (0.0%) |
| 32 (4.6%)[b] | failed SpO₂ measurement and not WHO clinically eligible[b] | 1 (3.1%)[b] | 32 (4.6%)[b] | failed SpO₂ measurement and not WHO clinically eligible[b] | 1 (3.1%)[b] |

[a]Hypoxaemic cases and deaths identified with pulse oximetry that would not have been identified using clinical guidelines alone.

[b]Cases and deaths identified by failure of attempted pulse oximetry that would not have been identified using clinical guidelines alone.

**Abbreviations:** CHW, community health worker; HC, health centre; SpO₂, oxygen saturation; WHO, World Health Organization.

where $\mu_i$ is the probability of death for individual $i$, $Y_1$ is the outcome, death, for individual $i$, and the exponent of $\beta_1$ is the modelled parameter of interest: the relative risk of death when SpO₂ is measured at <90% ($X_{1\_1} = 1$) compared to when it is measured at ≥90% ($X_{1\_1} = 0$). Children whose SpO₂ reading failed are separately categorised ($X_{1\_1} = 2$), not shown for

**Table 3. Independent associations of SpO$_2$ and danger sign exposures on mortality, unadjusted GLM regression results.**

**CHW Data**

| Model | Coefficient | RR | 95% CI | p-value |
|---|---|---|---|---|
| **M90. Malawi (= WHO) guidelines, <90% SpO$_2$ threshold (this was used by the healthcare workers) (N = 409)** | SpO$_2 \geq$ 90% | 1 (ref) | | |
| | <90% | 6.85 | (1.15–40.9) | 0.035 |
| | failed | 0.64 | (0.08–4.78) | 0.662 |
| | Malawi danger signs: absent | 1 (ref) | | |
| | present | 1.66 | (0.39–7.11) | 0.494 |
| | SpO$_2$ < 90% × danger signs | | (empty)[a] | |
| | failed SpO$_2$ × danger signs | | (empty)[a] | |
| | constant (baseline risk) | 0.036 | (0.021–0.064) | <0.001 |
| **M93. Malawi (= WHO) guidelines, <93% SpO$_2$ threshold (N = 412)** | SpO$_2 \geq$ 93% | 1 (ref) | | |
| | <93% | 3.00 | (0.44–20.7) | 0.264 |
| | failed | 0.63 | (0.84–4.71) | 0.651 |
| | Malawi danger signs: absent | 1 (ref) | | |
| | present | 0.87 | (0.12–6.48) | 0.893 |
| | SpO$_2$ <90% × danger signs | 2.07 | (0.81–52.9) | 0.661 |
| | failed SpO$_2$ × danger signs | | (empty)[b] | |
| | constant (baseline risk) | 0.037 | (0.021–0.065) | <0.001 |

**HC Data**

| Model | Coefficient | RR | 95% CI | p-value |
|---|---|---|---|---|
| **M90. Malawi guidelines, <90% SpO$_2$ threshold (this was used by the healthcare workers) (N = 646)** | SpO$_2 \geq$ 90% | 1 (ref) | | |
| | <90% | 5.73 | (1.68–19.5) | 0.005 |
| | failed | 10.8 | (1.02–113.1) | 0.048 |
| | Malawi danger signs: absent | 1 (ref) | | |
| | present | 4.05 | (0.75–21.9) | 0.104 |
| | SpO$_2$ <90% × danger signs | | (empty)[c] | |
| | failed SpO$_2$ × danger signs | | (empty)[c] | |
| | constant (baseline risk) | 0.005 | (0.001–0.021) | <0.001 |
| **M93. Malawi guidelines, <93% SpO$_2$ threshold (N = 622)** | SpO$_2 \geq$ 93% | 1 (ref) | | |
| | <93% | 3.99 | (1.06–15.0) | 0.041 |
| | failed | 10.1 | (0.96–106.1) | 0.054 |
| | Malawi danger signs: absent | 1 (ref) | | |
| | present | 3.58 | (0.60–21.2) | 0.160 |
| | SpO$_2$ < 90% × danger signs | | (empty)[c] | |
| | failed SpO$_2$ × danger signs | | (empty)[c] | |
| | constant (baseline risk) | 0.006 | (0.001–0.022) | <0.001 |
| **W90. WHO guidelines, <90% SpO$_2$ threshold (N = 675)** | SpO$_2 \geq$ 90% | 1 (ref) | | |
| | <90% | 9.37 | (2.17–40.4) | 0.003 |
| | failed | 4.00 | (0.46–34.8) | 0.209 |
| | WHO danger signs: absent | 1 (ref) | | |
| | present | 3.88 | (0.72–20.8) | 0.113 |
| | SpO$_2$ < 90% × danger signs | 0.44 | (0.04–4.23) | 0.478 |
| | failed SpO$_2$ × danger signs | | (empty)[d] | |
| | constant (baseline risk) | 0.008 | (0.003–0.021) | <0.001 |

*(Continued)*

**Table 3.** (Continued)

| W93. WHO guidelines, <93% SpO$_2$ threshold (N = 675) | SpO$_2 \geq$ 93% | 1 (ref) | | |
|---|---|---|---|---|
| | <93% | 6.68 | (1.52–29.4) | 0.012 |
| | failed | 4.80 | (0.51–44.9) | 0.169 |
| | WHO danger signs: absent | 1 (ref) | | |
| | present | 5.69 | (0.97–33.3) | 0.054 |
| | SpO$_2$ < 93% × danger signs | 0.34 | (0.03–3.30) | 0.350 |
| | failed SpO$_2$ × danger signs | | (empty)[d] | |
| | constant (baseline risk) | 0.007 | (0.002–0.020) | <0.001 |

× = interaction term. Please note that we know these models are correctly specified because they predict the observed mortality rates for each category shown in Table 2.

(empty) = no deaths in this group, so coefficient was not possible to estimate.

[a]See Table 2, CHW data, left orange panel, n = 3 and 0 deaths in group 'SpO$_2$ < 90% but Malawi clinically eligible' and n = 5 and 0 deaths in group 'failed SpO$_2$ measurement but Malawi clinically eligible'.

[b]See Table 2, CHW data, right yellow panel, n = 5 and 0 deaths in group 'failed SpO$_2$ measurement but Malawi clinically eligible'.

[c]See Table 2, HC data, top left orange panel and top right yellow panel, n = 34 and 0 deaths in group 'failed SpO$_2$ measurement but Malawi clinically eligible' and n = 15 (SpO$_2$ < 90%) or n = 39 (SpO$_2$ < 93%) and 0 deaths in group 'SpO$_2$ < 90% (<93%) only and not Malawi clinically eligible'.

[d]See Table 2, HC data, bottom left green panel and bottom right blue panel, n = 20 and 0 deaths in group 'failed SpO$_2$ measurement but WHO clinically eligible'.

**Abbreviations:** CHW, community health worker; GLM, generalised linear model; HC, health centre; ref, reference (baseline) category; RR, risk ratio; SpO$_2$, oxygen saturation; WHO, World Health Organization.

simplicity), controlling for presence of Malawi guidelines clinical signs ($X_{2\_1}$). We constructed separate models for SpO$_2$ < 93% ($X_{1\_2}$) (Model M93) and, for HC data for which WHO guidelines are different from Malawi guidelines, for WHO ($X_{2\_2}$) guidelines (Models W90 and W93; see Table 3).

We adjusted for confounding by age, sex, respiratory rate, and, in analyses of HC data only, Weight for Age Z-score (WAZ). Missing data prevented us from including maternal age, education, marital status, and wealth quintile as potential confounders. Too few deaths in each exposure group prevented assessment of effect modification by age group, sex, or CHW or HC level. Although we adjusted for clustering of illness episodes by child and CHW and HC catchment area, these models were unstable and not presented. Our unadjusted models were robust to clustering, with similar headline results following 100 iterations for models that did not converge.

The extent of missing data on outcomes due to the majority of outpatient episodes remaining unmatched to 30-day postdiagnosis survival (mortality) outcomes (Table 1) meant that multiple imputation of the missing outcome data was not feasible. The small numbers of deaths and episodes with low SpO$_2$ also precluded our planned regression discontinuity analyses of the effect of changing the SpO$_2$ threshold on mortality and referral outcomes.

Separately to the fatality outcome, we determined the association between our SpO$_2$/clinical sign exposures and hospital referral as the outcome ($Y_2$) using the same logistic regression Eq (1) except substituting the fatality outcome ($Y_1$) with $Y_2$. Because not all severely ill children referred to hospital actually arrive, as a sensitivity analysis, we repeated this analysis with referral decision from the outpatient exposure data set regardless of actual hospital arrival as the outcome ($Y_3$).

We calculated the sensitivity, specificity, and diagnostic odds ratio (DOR, with 95%CI) [21] of clinical and SpO$_2$ eligibility on the mortality outcome and compared it to the sensitivity, specificity, and DOR of clinical eligibility only for SpO$_2$ eligibility thresholds of <90% and <93% SpO$_2$ (WHO and Malawi eligibilities) at CHW and HC levels. We included SpO$_2$ failed

measurements as well as $SpO_2$ below threshold as $SpO_2$ eligible in these analyses, given failed $SpO_2$ measurements are associated with mortality.

## Ethics statement

Because data were deidentified and analysed anonymously, no authorisation or waiver of authorisation by patients for the release of individually identifiable protected health information was required. This study is a data linkage study, and the data it links together are from studies approved by the ethics boards of University College London (protocol 2006/002), the Malawi National Health Sciences Research Committee (protocols 941 and 837), and the London School of Hygiene & Tropical Medicine (protocol 6047), as detailed in the published research articles from the original studies [6, 9, 13–16].

## Results

We successfully matched 417/7,357 (5.7%) CHW diagnosed pneumonia episodes and 695/6,456 (10.8%) HC pneumonia episodes to outcome data (Fig 2). Sixteen (3.8%) of the 417 CHW patients and 13 (1.9%) of the 695 HC patients died within 7 days of diagnosis. There were no deaths between 8 and 29 days after the initial assessment. A further 9 CHW and 17 HC children were recorded as dying between 30 and 1,537 days after outpatient diagnosis and are included in the survival at 30 days groups. There were up to 7 pneumonia episodes in the same child in both CHW and HC data sets combined (mean: 1.2, SD: 0.6), with 1,112 episodes amongst 931 children, though only 36 children had multiple care-seeking episodes, and 32 of these had multiple episodes within 30 days of each other (S2 Appendix Table A). These episodes are all retained in the analysis given the relevance of each opportunity for case management. $SpO_2 < 90\%$ hypoxemia was more common in HC pneumonia episodes (608/6,459 = 9.4%) than CHW episodes (88/7,358 = 1.2%, Table 1).

There were some differences in exposures ($SpO_2$ measurements, chest indrawing, danger signs) and potential confounders (age, sex, respiratory rate, WAZ) between matched and unmatched episodes (Table 1). CHW patients with matched outcome data were less likely to have $SpO_2$ recorded and less likely to have saturations below 93% when recorded (though there was no significant difference in the proportions with saturations below 90%; Table 1). CHW patients with outcome data were less likely to have been recorded as vomiting everything and having convulsions, and consequently as having danger signs eligible for referral

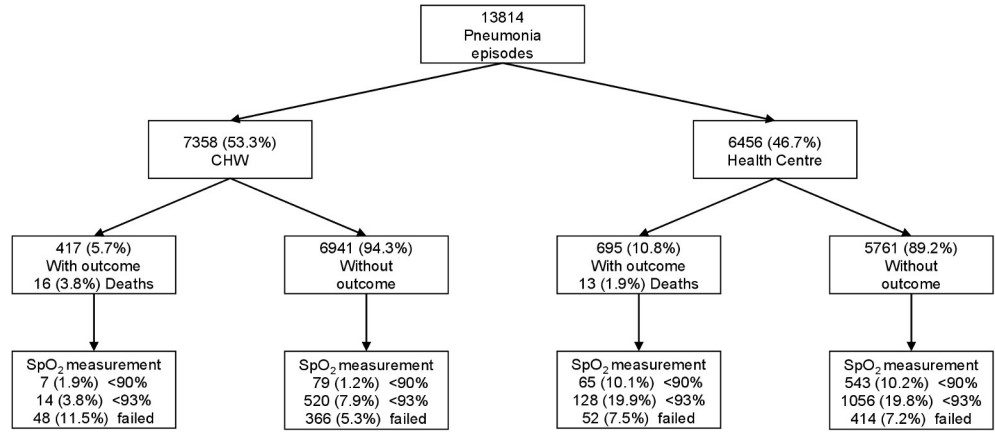

**Fig 2. Pneumonia episodes matched to outcome data.** CHW, community health worker; $SpO_2$, oxygen saturation.

according to WHO and Malawi guidelines (which are the same at CHW level), than CHW patients not matched to outcome data (Table 1), though they were more likely to have been recorded as having severe acute malnutrition. CHW patients with matched outcome data were also more likely to be male (53%) and younger (mean age 10 months old) than those not matched to outcome data (48% male, mean age 24 months old).

HC patients with matched outcome data were not found to have any significant differences in $SpO_2$ categories compared with those without matched outcome data (Table 1). HC patients with outcome data were more likely to have been recorded as HIV exposed/infected and of being apnoeic, and consequently (though there were no other significant differences across the danger signs) as having danger signs eligible for referral according to WHO guidelines (16%), than HC patients not matched to outcome data (11%; Table 1). HC patients with matched outcome data were more likely to have chest indrawing recorded (35%) than those unmatched to outcome data (25%) and consequently more likely to have Malawi guideline danger signs eligible for referral (40%) than those unmatched to outcome data (29%; Table 1). HC patients with matched outcome data were also more likely to be male (56%), were younger (mean age 8 months old) than those not matched to outcome data (52% male, mean age 16 months old), and were less likely to have low WAZs, on average (Table 1).

Table 2 CHW data show the distribution of the 16 deaths at CHW level and risk of death for each of the 2 $SpO_2$ and danger sign exposure group sets. Those who were not clinically or $SpO_2$ eligible for referral had lower mortality rates than those $SpO_2$-eligible only (3.6%–3.7% versus 11%–25%).

Table 2 HC data show the distribution of the 13 deaths at HC level and risk of death for each of the 4 $SpO_2$ and danger sign exposure group sets. In all 4 scenarios, those who are not clinically or $SpO_2$ eligible for referral had lower mortality rates than those both clinically and $SpO_2$ eligible (0.5%–0.8% versus 7.9%–12.5%). $SpO_2$-eligible only episodes had higher mortality than clinically eligible only episodes when applying WHO IMCI chart booklet criteria, especially using the <90% $SpO_2$ threshold (7.3% versus 3.0%).

For HC episodes, $SpO_2 < 90\%$ and <93%, respectively, identified 3 (23%) and 4 (31%) of the 13 deaths that would not have been identified using WHO IMCI guidelines alone (Table 2). Malawi guidelines, which include chest indrawing, identified all of these deaths, and 10 (77%) of the 13 HC deaths in total compared with 5 (38%) of the 13 HC deaths identified by WHO IMCI guidelines (Table 2).

For CHW episodes, $SpO_2 < 90\%$ and <93% both identified 1 (6%) of the 16 deaths that would not have been identified using iCCM/Malawi guidelines alone (Table 2). Twelve (75%) of the 16 deaths amongst CHW episodes were not identified by iCCM guidelines, which only identified 2 (12.5%) of the 16 deaths (Table 2). These 12 deaths also had $SpO_2 \geq 93\%$. Failed $SpO_2$ measurement independently identified 1 (6%) of the 16 CHW deaths and 1 (8%) of the 13 HC deaths (Table 2).

The unadjusted GLM analyses of the associations of $SpO_2$ and danger sign exposures with death predict the observed mortality rates for each exposure group. In CHW episodes (Table 3), there is an increased risk of death for those with $SpO_2 < 90\%$ than those with $SpO_2 \geq 90\%$, independent of the presence of danger signs (RR: 6.85, 95% CI: 1.15–40.9, p = 0.035). The $SpO_2 < 93\%$ threshold was not associated with an increased risk of death compared to those with $SpO_2 \geq 93\%$ (RR: 3.00, 95% CI: 0.44–20.7, p = 0.264)

In HC episodes, only 2–4 of 13 deaths occurred in the 'reference' $SpO_2 \geq 90\%/93\%$ no-danger–signs groups (Table 2), and both the $SpO_2 < 90\%$ (RR: 9.37 [95% CI: 2.17–40.4]) and <93% (RR: 6.68 [95% CI: 1.52–29.4]) thresholds are significantly associated with mortality in both the Malawi and WHO IMCI guideline models (Table 3).

Adjusting the models for age in months, sex, respiratory rate, and, for HC data, WAZ produced broadly similar results to the unadjusted models in Table 3 (S2 Appendix Table B and associated text).

Table 4 shows the outpatient referral decision indication by healthcare providers and linked hospital inpatient records for each of the $SpO_2$ and danger sign exposure sets. For CHWs, only 9.3% (39) of the 417 episodes had outpatient referral decisions, and these were more common in patients who were clinically or $SpO_2$ eligible (33.3%–100%) than those who were not (3.6%–7.0%). Only 0.7% (3) of the CHW episodes were found to be hospital inpatients within 7 days of outpatient diagnosis (Table 4), precluding regression analysis of this outcome in CHW episodes. In the HC data (Table 4), 30.4% (211) of the 695 patients had an outpatient referral decision. Notably, for Malawi guidelines, including <90% $SpO_2$ threshold (i.e., what was used by the healthcare workers in practice), healthcare workers referred 90.6% who were eligible by both criteria, 62.3% who were clinically eligible only, and only 20.0% of those eligible because of $SpO_2 < 90\%$ only. Seventy (10.1%) of the 695 HC episodes were hospital inpatients within 7 days; 60 of these followed an outpatient referral decision (28% of the 211 referral decisions were admitted as inpatients within 7 days).

Table 5 CHW data show both low $SpO_2$ and danger signs are associated with outpatient referral decisions for CHWs. Table 5 HC data show the same for HC episodes and that failed $SpO_2$ readings were also associated with referral decisions in WHO guideline models. Referrals were also associated with low $SpO_2$ under $SpO_2 < 90\%$ and <93% thresholds and danger signs according to both Malawi and WHO IMCI protocols in HC episodes (Table 5). The interaction terms between the $SpO_2$ and danger sign parameters were below 1 and statistically significant for the following HC referral models (Table 5): M90 and M93, outpatient referral decision indication outcome, $SpO_2 < 90\% \times$ danger signs; hospitalisation outcome, failed $SpO_2 \times$ danger signs; and W90, outpatient referral decision indication outcome, $SpO_2 < 90\% \times$ danger signs. These results indicate that the association between the $SpO_2$ parameter and the outcome was attenuated by the presence of danger signs. Hospital referrals and outpatient episodes with referral decision indications were associated with mortality (see S2 Appendix, p. 11 for further details).

Table 6 shows the sensitivity, specificity, and DOR of pulse oximetry with clinical signs versus clinical signs only in identifying deaths. We focus on sensitivity because the objective is to identify as many of the deaths as possible. At the CHW level, using both $SpO_2$ thresholds, the sensitivity was 25% compared with a sensitivity of 12.5% for Malawi clinical iCCM criteria only. At the HC level, both <90% and <93% $SpO_2$ eligibility thresholds give a sensitivity of 85% compared with a sensitivity of 77% for Malawi clinical IMCI criteria only. At the HC level, in scenarios using WHO clinical criteria, 9 of the 13 deaths are identified by the <90% $SpO_2$ eligibility threshold, and 10 of the 13 deaths are identified by the <93% $SpO_2$ eligibility thresholds. These sensitivities of 69% and 77% compare with a sensitivity of 38% for WHO clinical IMCI criteria only. Given the small numbers of deaths, the differences in sensitivity between the scenarios are not statistically significant. The DORs show that at the CHW level, neither pulse oximetry with clinical signs (Table 6: <90% $SpO_2$ threshold, DOR 1.26 [95% CI: 0.40–4.00] and <93% $SpO_2$ threshold, DOR 1.17 [95% CI: 0.37–3.71]) nor iCCM clinical signs alone (DOR: 1.33 [95% CI: 0.29–6.05]) accurately identified those who died. At the HC level, pulse oximetry was able to accurately identify those who die with both Malawi (<90% $SpO_2$ threshold, DOR: 7.13 [95% CI: 1.57–32.4]; <93% $SpO_2$ threshold, DOR: 6.19 [95% CI: 1.36–28.1]) and WHO (<90% $SpO_2$ threshold, DOR: 6.57 [95% CI: 2.00–21.6]; <93% $SpO_2$ threshold, DOR: 6.82 [95% CI: 1.86–25.0]) IMCI clinical signs. Although the point estimates for these DORs are higher than those for when Malawi and WHO IMCI clinical signs alone are used without pulse oximetry (Malawi: DOR 5.25 [95% CI: 1.43–19.2], WHO: DOR 3.43 [95%

**Table 4. Outpatient referral decision indication and hospital inpatients within 7 days by SpO₂ and danger sign exposure group sets.**

**CHW Data**

| N = 417, n (%) | Malawi guidelines (= WHO guidelines), <90% SpO$_2$ threshold (this was used by the healthcare workers) | Outpatient referral decision, n (%) | Hospital inpatients within 7 days, n (%) | N = 417, n (%) | Malawi guidelines (= WHO guidelines), <93% SpO$_2$ threshold | Outpatient referral decision, n (%) | Hospital inpatients within 7 days, n (%) |
|---|---|---|---|---|---|---|---|
| 329 (78.9%) | NOT Malawi clinically eligible and SpO$_2$ > 90% | 12 (3.6%) | 1 (0.3%) | 324 (77.7%) | NOT Malawi clinically eligible and SpO$_2$ ≥ 93% | 12 (3.7%) | 1 (0.3%) |
| 33 (7.9%) | Malawi clinically eligible only and SpO$_2$ > 90% | 15 (45.5%) | 1 (3.0%) | 31 (7.4%) | Malawi clinically eligible only and SpO$_2$ ≥ 93% | 13 (41.9%) | 1 (3.2%) |
| 4 (1.0%)[a] | SpO$_2$ < 90% only and not Malawi clinically eligible[a] | 3 (75.0%)[a] | 0 (0.0%)[a] | 9 (2.2%)[a] | SpO$_2$ <93% only and not Malawi clinically eligible[a] | 3 (33.3%)[a] | 0 (0.0%)[a] |
| 3 (0.7%) | SpO$_2$ < 90% and Malawi clinically eligible | 3 (100%) | 1 (33.3%) | 5 (1.2%) | SpO$_2$ < 93% and Malawi clinically eligible | 5 (100%) | 1 (20.0%) |
| 5 (1.2%) | failed SpO$_2$ measurement but Malawi clinically eligible | 3 (60.0%) | 0 (0.0%) | 5 (1.2%) | failed SpO$_2$ measurement but Malawi clinically eligible | 3 (60.0%) | 0 (0.0%) |
| 43 (10.3%)[b] | failed SpO$_2$ measurement and not Malawi clinically eligible[b] | 3 (7.0%)[b] | 0 (0.0%)[b] | 43 (10.3%)[b] | failed SpO$_2$ measurement and not Malawi clinically eligible[b] | 3 (7.0%)[b] | 0 (0.0%)[b] |

**HC Data**

| N = 695, n (%) | Malawi guidelines, <90% SpO$_2$ threshold (this was used by the healthcare workers) | Outpatient referral decision, n (%) | Hospital inpatients within 7 days, n (%) | N = 695, n (%) | Malawi guidelines, <93% SpO$_2$ threshold | Outpatient referral decision, n (%) | Hospital inpatients within 7 days, n (%) |
|---|---|---|---|---|---|---|---|
| 387 (55.7%) | NOT Malawi clinically eligible and SpO$_2$ > 90% | 12 (3.1%) | 7 (1.8%) | 363 (52.2%) | NOT Malawi clinically eligible and SpO$_2$ ≥ 93% | 9 (2.5%) | 5 (1.4%) |
| 191 (27.5%) | Malawi clinically eligible only and SpO$_2$ > 90% | 119 (62.3%) | 39 (20.4%) | 152 (21.9%) | Malawi clinically eligible only and SpO$_2$ ≥ 93% | 86 (56.6%) | 27 (17.8%) |
| 15 (2.2%)[a] | SpO$_2$ < 90% only and not Malawi clinically eligible[a] | 3 (20.0%)[a] | 0 (0.0%)[a] | 39 (5.6%)[a] | SpO$_2$ < 93% only and not Malawi clinically eligible[a] | 6 (15.4%)[a] | 2 (5.1%)[a] |
| 50 (7.2%) | SpO$_2$ < 90% and Malawi clinically eligible | 45 (90.0%) | 16 (32.0%) | 89 (12.8%) | SpO$_2$ < 93% and Malawi clinically eligible | 78 (87.6%) | 28 (31.5%) |
| 34 (4.9%) | failed SpO$_2$ measurement but Malawi clinically eligible | 31 (91.2%) | 6 (17.6%) | 34 (4.9%) | failed SpO$_2$ measurement but Malawi clinically eligible | 31 (91.2%) | 6 (17.6%) |
| 18 (2.6%)[b] | failed SpO$_2$ measurement and not Malawi clinically eligible[b] | 1 (5.6%)[b] | 2 (11.1%)[b] | 18 (2.6%)[b] | failed SpO$_2$ measurement and not Malawi clinically eligible[b] | 1 (5.6%)[b] | 2 (11.1%)[b] |

| N = 695, n (%) | WHO guidelines, <90% SpO$_2$ threshold | Outpatient referral decision, n (%) | Hospital inpatients within 7 days, n (%) | N = 695, n (%) | WHO guidelines, <93% SpO$_2$ threshold | Outpatient referral decision, n (%) | Hospital inpatients within 7 days, n (%) |
|---|---|---|---|---|---|---|---|
| 512 (73.7%) | Not WHO clinically eligible and SpO$_2$ ≥ 90% | 101 (19.7%) | 33 (6.4%) | 461 (66.3%) | Not WHO clinically eligible and SpO$_2$ ≥ 93% | 74 (16.1%) | 23 (5.0%) |
| 66 (9.5%) | WHO clinically eligible only and SpO$_2$ ≥ 90% | 30 (45.5%) | 13 (19.7%) | 54 (7.8%) | WHO clinically eligible only and SpO$_2$ ≥ 93% | 21 (38.9%) | 9 (16.7%) |
| 41 (5.9%)[a] | SpO$_2$ < 90% only and not WHO clinically eligible[a] | 27 (65.9%)[a] | 9 (22.0%)[a] | 92 (13.2%)[a] | SpO$_2$ < 93% only and not WHO clinically eligible[a] | 54 (58.7%)[a] | 19 (20.7%)[a] |
| 24 (3.5%) | SpO$_2$ < 90% and WHO clinically eligible | 21 (87.5%) | 7 (29.2%) | 36 (5.2%) | SpO$_2$ < 93% and WHO clinically eligible | 30 (83.3%) | 11 (30.6%) |
| 20 (2.9%) | failed SpO$_2$ measurement but WHO clinically eligible | 18 (90.0%) | 4 (20.0%) | 20 (2.9%) | failed SpO$_2$ measurement but WHO clinically eligible | 18 (90.0%) | 4 (20.0%) |

(*Continued*)

**Table 4.** (Continued)

| 32 (4.6%)[b] | failed SpO$_2$ measurement and not WHO clinically eligible[b] | 14 (43.8%)[b] | 4 (12.5%)[b] | 32 (4.6%)[b] | failed SpO$_2$ measurement and not WHO clinically eligible[b] | 14 (43.8%)[b] | 4 (12.5%)[b] |
|---|---|---|---|---|---|---|---|

[a]Hypoxaemic episodes identified with pulse oximetry that would not have been identified using clinical guidelines alone.

[b]Episodes identified by failure of attempted pulse oximetry that would not have been identified using clinical guidelines alone.

**Abbreviations:** CHW, community health worker; HC, health centre; SpO$_2$, oxygen saturation; WHO, World Health Organization.

CI: 1.10–10.7]), the confidence intervals are wide and overlapping, indicating these differences are not statistically significant.

## Discussion

We show the important role of confirmed hypoxaemia in identifying otherwise unrecognised fatal childhood pneumonia episodes accessing primary healthcare centres by applying the 2014 WHO IMCI clinical signs, which do not include chest indrawing as a referral sign, in children aged 2–59 months old. Pulse oximeter use will identify hypoxaemia in up to 30% of episodes that are not identified by clinical signs alone [22]. The 2014 WHO IMCI recommends use of pulse oximetry when available and referral to the hospital if SpO$_2$ is less than 90%; however, pulse oximetry is not frequently available at the outpatient level. With low-cost portable pulse oximeters becoming much more available compared to 2014, especially in the context of the ongoing COVID pandemic [23], it is likely that they will be available more widely and used more often at the outpatient level than before. Even so, it will further facilitate pulse oximetry use if WHO were to recommend them without any conditions to evaluate pneumonia episodes.

Pre-2018 Malawi guidelines for community and outpatient case management [17] were consistent with the previous WHO guidelines [19, 24], i.e., chest indrawing is considered a referral criteria. This criteria was removed as a referral sign in the revised 2014 WHO IMCI chart booklet [8] following controlled low-mortality trials that indicated chest indrawing episodes could be managed at home without increased risk [25, 26], and Malawi made this change too in 2018. Since then, 3 other randomised trials—one each in India, Kenya, and Malawi—have demonstrated that chest indrawing pneumonia can be treated with oral amoxicillin on an outpatient basis [27–29]. However, for CHWs, chest indrawing remains a referral sign. The pre-2018 Malawi IMCI protocol identified 2-fold more (10/13 = 77%) of the HC patients who died within 7 days of outpatient diagnosis than the 2014 WHO IMCI protocol (5/13 = 38%). These data and recent data from hospitalised children in Kenya showed that in routine care settings, presence of chest indrawing may require referral in some high-mortality settings [30]. Because not much is known about other factors that influenced adverse outcomes in these children with chest indrawing, such as care seeking and appropriate and timely treatment, further research is required to examine the beneficial effects of hospitalisation of children with chest indrawing in routine settings.

An abnormal SpO$_2$ measured by pulse oximetry most commonly indicates ventilation–perfusion mismatch from a pulmonary illness and is considered a more objective measurement than observing chest indrawing, which can be subjective and hard to see [31]. Therefore, although chest indrawing identified all of the deaths from HC episodes and all but one of the deaths from CHW episodes identified by pulse oximetry, pulse oximetry may lead to more reliable decision-making. However, pulse oximetry can also be done poorly, has cost implications, and still needs to be conducted in conjunction with a thorough clinical assessment.

**Table 5. Independent associations of SpO$_2$ and danger sign exposures on referrals, unadjusted GLM regression results.**

**CHW Data**

| Model | Coefficient | RR | (95% CI) | p-value |
|---|---|---|---|---|
| | | colspan Outcome = outpatient referral decision indication | | |

| Model | Coefficient | RR | (95% CI) | p-value |
|---|---|---|---|---|
| **M90. Malawi (= WHO) guidelines, <90% SpO$_2$ threshold (this was used by the healthcare workers) (N = 409)†** | SpO$_2 \geq$ 90% | 1 (ref) | | |
| | <90% | 79.3 | (7.67–819.0) | 0.002 |
| | failed | 1.98 | (0.54–7.3) | 0.305 |
| | Malawi danger signs: absent | 1 (ref) | | |
| | present | 22.0 | (8.99–53.9) | <0.001 |
| | SpO$_2$ < 90% × danger signs | | (empty)[a] | |
| | failed SpO$_2$ × danger signs | 0.91 | (0.09–9.24) | 0.935 |
| | constant (baseline risk) | 0.038 | (0.021–0.067) | <0.001 |
| **M93. Malawi (= WHO) guidelines, <93% SpO$_2$ threshold (N = 412)** | SpO$_2 \geq$ 93% | 1 (ref) | | |
| | <93% | 9.00 | (3.06–26.5) | <0.001 |
| | failed | 1.88 | (0.55–6.41) | 0.311 |
| | Malawi danger signs: absent | 1 (ref) | | |
| | present | 11.3 | (5.66–22.6) | <0.001 |
| | SpO$_2$ < 93% × danger signs | | (empty)[b] | |
| | failed SpO$_2$ × danger signs | 0.76 | (0.17–3.32) | 0.715 |
| | constant (baseline risk) | 0.037 | (0.021–0.064) | <0.001 |

**HC Data**

| Model | Coefficient | Outcome = outpatient referral decision indication | | | Outcome = hospitalisation (within 7 days) | | |
|---|---|---|---|---|---|---|---|
| | | RR | (95% CI) | p-value | RR | (95% CI) | p-value |
| **M90. Malawi guidelines, <90% SpO$_2$ threshold (this was used by the health workers) (left: N = 695; right: N = 680)** | SpO$_2 \geq$ 90% | 1 (ref) | | | 1 (ref) | | |
| | <90% | 6.45 | (2.03–20.5) | 0.002 | 1.56 | (0.96–2.56) | 0.073 |
| | failed | 1.79 | (0.25–13.0) | 0.565 | 6.14 | (1.37–27.5) | 0.018 |
| | Malawi danger signs: absent | 1 (ref) | | | 1 (ref) | | |
| | present | 20.09 | (11.4–35.5) | <0.001 | 11.3 | (5.15–24.8) | <0.001 |
| | SpO$_2$ < 90% × danger signs | 0.22 | (0.07–0.72) | 0.012 | | (empty)[c] | |
| | failed SpO$_2$ × danger signs | 0.82 | (0.11–5.98) | 0.842 | 0.14 | (0.03–0.76) | 0.023 |
| | constant (baseline risk) | 0.031 | (0.018–0.054) | 0.000 | 0.018 | (0.009–0.038) | <0.001 |
| **M93. Malawi guidelines, <93% SpO$_2$ threshold (N = 695)** | SpO$_2 \geq$ 93% | 1 (ref) | | | 1 (ref) | | |
| | <93% | 6.21 | (2.33–16.5) | <0.001 | 3.72 | (0.75–18.6) | 0.109 |
| | failed | 2.24 | (0.30–16.7) | 0.432 | 8.07 | (1.68–38.8) | 0.009 |
| | Malawi danger signs: absent | 1 (ref) | | | 1 (ref) | | |
| | present | 22.8 | (11.8–44.2) | <0.001 | 12.9 | (5.06–32.9) | <0.001 |
| | SpO$_2$ < 93% × danger signs | 0.25 | (0.09–0.67) | 0.006 | 0.48 | (0.09–2.53) | 0.383 |
| | failed SpO$_2$ × danger signs | 0.72 | (0.10–5.41) | 0.749 | 0.12 | (0.02–0.72) | 0.020 |
| | constant (baseline risk) | 0.025 | (0.013–0.047) | <0.001 | 0.014 | (0.006–0.033) | <0.001 |

*(Continued)*

**Table 5.** (Continued)

| | | | | | | | |
|---|---|---|---|---|---|---|---|
| **W90. WHO guidelines, <90% SpO₂ threshold (N = 695)** | $SpO_2 \geq 90\%$ | 1 (ref) | | | 1 (ref) | | |
| | <90% | 3.34 | (2.52–4.42) | <0.001 | 3.41 | (1.75–6.62) | <0.001 |
| | failed | 2.22 | (1.44–3.41) | <0.001 | 1.94 | (0.73–5.14) | 0.183 |
| | WHO danger signs: absent | 1 (ref) | | | 1 (ref) | | |
| | present | 2.30 | (1.68–3.16) | <0.001 | 3.06 | (1.70–5.50) | <0.001 |
| | $SpO_2 < 90\% \times$ danger signs | 0.58 | (0.38–0.87) | 0.009 | 0.43 | (0.15–1.22) | 0.114 |
| | failed $SpO_2 \times$ danger signs | 0.89 | (0.53–1.51) | 0.672 | 0.52 | (0.13–2.12) | 0.364 |
| | constant (baseline risk) | 0.197 | (0.166–0.235) | <0.001 | 0.064 | (0.046–0.090) | <0.001 |
| **W93. WHO guidelines, <93% SpO₂ threshold (N = 695)†** | $SpO_2 \geq 93\%$ | 1 (ref) | | | 1 (ref) | | |
| | <93% | 7.43 | (4.58–12.1) | <0.001 | 4.14 | (2.35–7.28) | <0.001 |
| | failed | 4.07 | (1.94–8.54) | <0.001 | 2.51 | (0.92–6.81) | 0.072 |
| | WHO danger signs: absent | 1 (ref) | | | 1 (ref) | | |
| | present | 3.33 | (1.82–6.07) | <0.001 | 3.34 | (1.63–6.84) | 0.001 |
| | $SpO_2 < 93\% \times$ danger signs | 1.06 | (0.34–3.31) | 0.924 | 0.44 | (0.17–1.15) | 0.096 |
| | failed $SpO_2 \times$ danger signs | 3.47 | (0.62–19.6) | 0.157 | 0.48 | (0.11–2.06) | 0.322 |
| | constant (baseline risk) | 0.191 | (0.149–0.245) | <0.001 | 0.050 | (0.033–0.074) | <0.001 |

(empty) = no referrals in this group, so coefficient was not possible to estimate

× = interaction term. Please note that we know these models are correctly specified because they predict the observed referral rates for each category shown in Table 4.

†The CHW M90 and HC W93 outpatient referral decision GLMs with binomial family and log link did not converge; therefore, we report the analogous logistic regression models for these 2 analyses. These models report results in ORs rather than RRs. The ORs are more extreme than the RRs, especially for the HC outpatient referral decision outcome, which is relatively common (30%: 211 of 695 episodes; the outpatient referral decision is less common for CHW episodes: 9%: 39 out of 417 episodes).

[a]See Table 4 CHW data, top left orange panel, n = 4 and 0 referrals in group 'SpO₂ < 90% only and not Malawi clinically eligible'.

[b]See Table 4, CHW data, top right yellow panel, n = 9 and 0 referrals in group 'SpO₂ < 93% only and not Malawi clinically eligible'.

[c]See Table 4, HC data, orange panel, n = 15 and 0 referrals in group 'SpO₂ < 90% only and not Malawi clinically eligible'.

**Abbreviations:** CHW, community health worker; GLM, generalised linear model; HC, health centre; OR, Odds Ratio; ref, reference (baseline) category; RR, Risk Ratio; SpO₂, oxygen saturation; WHO, World Health Organization.

Additionally, a 'normal' SpO₂ reading should be considered an adjunct to, not in lieu of, recognition of clinical danger signs such as inability to drink when making decisions about hospital referral. Whilst pulse oximetry in conjunction with clinical signs can identify more fatal pneumonia episodes than clinical signs alone, this increased sensitivity of diagnosis comes at the price of reduced specificity (Table 6) and therefore has potential to overwhelm weak hospital systems via identifying additional patients for referral to hospital. Detailed understanding of local hospital system capacities will be fundamental when deciding the appropriate SpO₂ threshold for referring patients. Ultimately, further research is needed to examine these trade-offs, though our research indicates that both pulse oximetry and chest indrawing are important in a high-child–mortality setting.

Notably, 12 (75%) of the 16 deaths in CHW episodes that occurred within 7 days of diagnosis were neither hypoxaemic nor clinically eligible for referral highlighting the importance of follow-up and continuing case management, as well as more specific diagnostic approaches (S2 Appendix, pp. 11–12 has further detail on these episodes). CHWs may also have missed danger signs, meaning support for training, supportive supervision, and mentoring for iCCM may be important in this context [32]. The use of adult probes for pulse oximetry in our study may have contributed to missed hypoxaemic episodes. The increasing availability of paediatric probes may provide greater sensitivity to detect children who may die of pneumonia.

**Table 6. Sensitivity and specificity of pulse oximetry with clinical signs versus clinical signs only in identifying patients who die.**

**CHW Data**

| | Died | Alive | PPV | NPV | DOR (95% CI) | Sensitivity | Specificity | | Died | Alive | PPV | NPV | DOR (95% CI) | Sensitivity | Specificity |
|---|---|---|---|---|---|---|---|---|---|---|---|---|---|---|---|
| iCCM severe (including hypoxaemia <90% & failed SpO2) | 4 | 84 | 5% | 96% | 1.26 (0.40–4.00) | 25% | 79% | iCCM severe (including hypoxaemia <93%) | 4 | 89 | 4% | 96% | 1.17 (0.37–3.71) | 25% | 78% |
| iCCM nonsevere | 12 | 317 | | | | | | iCCM nonsevere | 12 | 312 | | | | | |
| versus | | | | | | | | | | | | | | | |
| iCCM severe | 2 | 39 | 5% | 96% | 1.26 (0.40–4.00) | 12.5% | 90% | | | | | | | | |
| iCCM nonsevere | 14 | 362 | | | | | | | | | | | | | |

**HC Data**

| | Died | Alive | PPV | NPV | DOR (95% CI) | Sensitivity | Specificity | | Died | Alive | PPV | NPV | DOR (95% CI) | Sensitivity | Specificity |
|---|---|---|---|---|---|---|---|---|---|---|---|---|---|---|---|
| Malawi IMCI severe (including hypoxaemia <90%) | 11 | 297 | 4% | 99% | 7.13 (1.57–32.4) | 85% | 56% | Malawi IMCI severe (including hypoxaemia <93%) | 11 | 321 | 3% | 99% | 6.19 (1.36–28.1) | 85% | 53% |
| IMCI nonsevere | 2 | 385 | | | | | | IMCI nonsevere | 2 | 361 | | | | | |
| versus | | | | | | | | | | | | | | | |
| Malawi IMCI severe | 10 | 265 | 4% | 99% | 5.25 (1.43–19.2) | 77% | 61% | | | | | | | | |
| Malawi IMCI nonsevere | 3 | 417 | | | | | | | | | | | | | |

| | Died | Alive | PPV | NPV | DOR (95% CI) | Sensitivity | Specificity | | Died | Alive | PPV | NPV | DOR (95% CI) | Sensitivity | Specificity |
|---|---|---|---|---|---|---|---|---|---|---|---|---|---|---|---|
| WHO IMCI severe (including hypoxaemia <90%) | 9 | 174 | 5% | 99% | 6.57 (2.00–21.6) | 69% | 74% | WHO IMCI severe (including hypoxaemia <93%) | 10 | 224 | 4% | 99% | 6.82 (1.86–25.0) | 77% | 67% |
| IMCI nonsevere | 4 | 508 | | | | | | IMCI nonsevere | 3 | 458 | | | | | |
| versus | | | | | | | | | | | | | | | |
| WHO IMCI severe | 5 | 105 | 5% | 99% | 3.43 (1.10–10.7) | 38% | 85% | | | | | | | | |
| WHO IMCI nonsevere | 8 | 577 | | | | | | | | | | | | | |

Alive denotes 30-day survival. DORs over 1 discriminate properly, i.e., those who have the feature are more likely to have the outcome. **Abbreviations:** CHW, community health worker; DOR, diagnostic odds ratio; HC, health centre; iCCM, integrated community case management; IMCI, integrated management of childhood illness; NPV, Negative Predictive Value; PPV, Positive Predictive Value; WHO, World Health Organization.

The CHW and HC workers in our study were instructed to follow the Malawi guidelines and <90% SpO$_2$ threshold for identifying outpatient episodes for referral. Nevertheless, we found that SpO$_2$ readings of <90% and <93% and, at the HC level, both WHO and Malawi clinical danger signs were associated with a healthcare provider decision to refer the child and hospitalisation within 7 days of outpatient diagnosis. Though most of the hospitalisations followed an outpatient referral decision, these represent only 2 (5%) of 39 episodes with an outpatient referral decision at the CHW level and 60 (28%) out of 211 with an outpatient referral decision at HC level. This is likely to be due to a combination of difficulties with transport,

finances or family circumstances precluding travel to hospital, or hospitals not admitting referred patients [33, 34]. It could also be due to incomplete hospital inpatient records and issues with matching outpatient episodes to inpatient records, preventing us from knowing of some successful hospitalisations (e.g., referrals to hospitals outside of our study area). When referral is not feasible, prompt treatment at the CHW and HC level with oral antibiotics is required as minimum care, pending injectable antibiotics and oxygen therapy as required.

Outpatient referral decision, and hospitalisations within 7 days of outpatient diagnosis were associated with mortality. All 12 deaths amongst the 73 hospital referrals occurred on the same day of hospitalisation and were from HC episodes (only 3 of the 73 hospitalisations were from CHW episodes). Nine of the 12 hospital deaths were on the same day as outpatient diagnosis and one the following day, suggesting rapid deterioration, delayed care seeking and presentation at the HC, or a combination of all three [4, 35].

At the CHW level, 13 (81%) of the 16 deaths were in patients for whom outpatient HC referral was not recommended, generally reflecting correct guideline application; i.e., the CHW did not record any clinical signs indicating they should be referred. Of these 13 deaths, none were subsequently hospitalised, and all died within 1–5 days of diagnosis by a CHW. This suggests they died at home whilst receiving oral antibiotic treatment. At the HC level, 4 (31%) of the 13 deaths were in episodes not referred to the hospital, although only one met referral criteria. All of these 4 deaths were eventually hospitalised. These episodes all present missed opportunities for intervention at outpatient primary care.

Our findings on referral decision-making are similar to those we previously reported using the whole CHW and HC data sets [6], suggesting our matched sample is representative of the wider sample in this respect. Our finding that significantly more patients with both clinical and $SpO_2$ eligibility were referred than those who were either only clinically or $SpO_2$ eligible suggests healthcare workers at the outpatient level considered both pulse oximetry and clinical signs for referral decision-making rather than one or the other. More research to understand how referral decisions are determined when pulse oximeters are available is needed. Training is also required to re-emphasise the importance of following both clinical guidelines and $SpO_2$ when making decisions to refer patients. Devices with in-built decision support or that are complemented by mHealth tools could play a role in supporting appropriate case management and should be evaluated for both clinical outcomes and changes in care seeking and referral outcomes.

Our main limitation was our ability to only match 6% of CHW episodes and 11% of HC episodes to mortality outcome data. Given the geographical coverage and age differences of the data sets, we were able to match 29% of the CHW episodes and 24% of the HC episodes for which matching was potentially feasible (S2 Appendix, page 3). Although the samples of CHW and HC episodes matched to mortality outcome data had some differences in characteristics from those unmatched to mortality outcome data, the characteristics of our sample and our findings are generalisable to similar LMIC populations likely to be found in high-mortality settings. The matched patients were younger on average than the unmatched patients; the mortality surveillance data only included children born after October 2011, and therefore, older children were only present in the clinical data set. The lower proportions with danger signs and $SpO_2 < 93\%$ in the matched CHW data compared with the unmatched CHW data suggest that the matched CHW patients were less sick on average, and therefore, we may have missed some deaths. On the other hand, HC matched patients were sicker on average than the unmatched HC patients, with greater proportions having danger signs, although there were no significant differences in $SpO_2$ categories (Table 1).

Given the similarity of the unadjusted and adjusted results, the fact the regression models already lack precision due to the small numbers of deaths, and that we were only able to use

age, sex, and respiratory rate as confounders, any propensity score matching analysis to balance such potential measured confounders across exposure groups was deemed futile and abandoned. Additionally, given the low matching rates, multiple imputation of missing data was not included because it produced imputations, and consequent regression coefficient estimates, that were too unstable. Analyses of effect modification were also not possible because of the small number of deaths in each category.

It is estimated that the implementation of IMCI enhanced with pulse oximetry may result in 103,000–145,000 pneumonia deaths averted in the 15 highest-burden countries [36], though this modelling study was based on assumptions about associations with mortality [36]. To help place our results in context and aid policy and decision-making, a cost-effectiveness analysis and economic evaluation of outpatient pulse oximetry—based on our results and cost, benefit (disability adjusted life years averted), and population parameters—is forthcoming. Further research looking at an approach in which chest indrawing episodes without other danger signs are referred or not solely based on $SpO_2$ is also warranted, as is research to evaluate the optimum $SpO_2$ threshold for action. Similar analyses evaluating pulse oximetry in low-mortality LMICs are necessary.

Pulse oximetry use by health workers at HCs can identify hypoxaemic pneumonia patients who go on to die who would otherwise be missed by current referral guidelines, especially if they do not include chest indrawing and pulse oximeters are unavailable. With low-cost portable pulse oximeters becoming much more available now compared to 2014, the WHO/UNICEF IMCI protocol could recommend pulse oximeters for pneumonia case management, instead of making a conditional recommendation of its use when available. As pulse oximetry use increases in outpatient settings, critical next steps include addressing oxygen availability at clinics and during transportation to hospitals. Our analysis also indicates that timely and appropriate care seeking by families and prompt management by health workers is key to survival of children with pneumonia.

## Supporting information

**S1 STROBE Checklist. STROBE checklist for reporting of observational studies.** STROBE, Strengthening the Reporting of Observational Studies in Epidemiology.
(DOCX)

**S1 Appendix. Prespecified analysis plan.**
(DOCX)

**S2 Appendix. Additional supporting information and results referred to in the manuscript.**
(DOCX)

## Author Contributions

**Conceptualization:** Tim Colbourn, Carina King, James Beard, Eric D. McCollum.

**Data curation:** Tim Colbourn, Carina King, James Beard.

**Formal analysis:** Tim Colbourn, Carina King, James Beard, Eric D. McCollum.

**Funding acquisition:** Tim Colbourn, Anthony Costello, Bejoy Nambiar, Shamim Ahmad Qazi, Yasir Bin Nisar, Eric D. McCollum.

**Investigation:** Tim Colbourn, Carina King, James Beard, Tambosi Phiri, Malizani Mdala, Beatiwel Zadutsa, Charles Makwenda, Anthony Costello, Norman Lufesi, Charles

Mwansambo, Bejoy Nambiar, Shubhada Hooli, Neil French, Naor Bar Zeev, Shamim Ahmad Qazi, Yasir Bin Nisar, Eric D. McCollum.

**Methodology:** Tim Colbourn, Carina King, James Beard, Shubhada Hooli, Neil French, Naor Bar Zeev, Shamim Ahmad Qazi, Yasir Bin Nisar, Eric D. McCollum.

**Project administration:** Tim Colbourn, Carina King, James Beard, Tambosi Phiri, Malizani Mdala, Beatiwel Zadutsa, Charles Makwenda, Shamim Ahmad Qazi, Yasir Bin Nisar, Eric D. McCollum.

**Supervision:** Tim Colbourn, Carina King, James Beard, Tambosi Phiri, Malizani Mdala, Beatiwel Zadutsa, Charles Makwenda, Eric D. McCollum.

**Validation:** Tim Colbourn, Carina King, James Beard, Tambosi Phiri, Malizani Mdala, Beatiwel Zadutsa, Charles Makwenda, Anthony Costello, Norman Lufesi, Charles Mwansambo, Bejoy Nambiar, Shubhada Hooli, Neil French, Naor Bar Zeev, Shamim Ahmad Qazi, Yasir Bin Nisar, Eric D. McCollum.

**Visualization:** Tim Colbourn.

**Writing – original draft:** Tim Colbourn, James Beard.

**Writing – review & editing:** Tim Colbourn, Carina King, James Beard, Tambosi Phiri, Malizani Mdala, Beatiwel Zadutsa, Charles Makwenda, Anthony Costello, Norman Lufesi, Charles Mwansambo, Bejoy Nambiar, Shubhada Hooli, Neil French, Naor Bar Zeev, Shamim Ahmad Qazi, Yasir Bin Nisar, Eric D. McCollum.

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
