## [Editor Report · Decision Letter 0]

17 Feb 2020

Dear Dr Colbourn, 

Thank you for submitting your manuscript entitled "Predictive value of pulse oximetry for mortality in infants and children presenting to primary care with clinical pneumonia in rural Malawi" for consideration by PLOS Medicine.

Your manuscript has now been evaluated by the PLOS Medicine editorial staff and I am writing to let you know that we would like to send your submission out for external peer review.

Kind regards,

Helen Howard, for Clare Stone PhD 

Acting Editor-in-Chief

PLOS Medicine 

plosmedicine.org

---

## [Decision Letter · Decision Letter 1]

19 Mar 2020

Dear Dr. Colbourn,

Thank you very much for submitting your manuscript "Predictive value of pulse oximetry for mortality in infants and children presenting to primary care with clinical pneumonia in rural Malawi" (PMEDICINE-D-20-00470R1) for consideration at PLOS Medicine. 

[LINK]

In light of these reviews, I am afraid that we will not be able to accept the manuscript for publication in the journal in its current form, but we would like to consider a revised version that addresses the reviewers' and editors' comments. Obviously we cannot make any decision about publication until we have seen the revised manuscript and your response, and we plan to seek re-review by one or more of the reviewers. 

We expect to receive your revised manuscript by Apr 09 2020 11:59PM. Please email us (plosmedicine@plos.org) if you have any questions or concerns.

We look forward to receiving your revised manuscript. 

Sincerely,

Adya Misra, PhD

Senior Editor 

PLOS Medicine

plosmedicine.org

Title: Please revise your title according to PLOS Medicine's style. Your title must be nondeclarative and not a question. It should begin with main concept if possible. "Effect of" should be used only if causality can be inferred, i.e., for an RCT. Please place the study design ("A randomized controlled trial," "A retrospective study," "A modelling study," etc.) in the subtitle (ie, after a colon).

Abstract- Please structure your abstract using the PLOS Medicine headings (Background, Methods and Findings, Conclusions). Funding is not required here. 

Abstract methods and findings- please provide demographics, places where this study took place in Malawi along with dates

Abstract Methods and Findings:

* Please ensure that all numbers presented in the abstract are present and identical to numbers presented in the main manuscript text.

* Please include the study design, population and setting, number of participants, years during which the study took place, length of follow up, and main outcome measures.

* Please quantify the main results (with 95% CIs and p values).

* Please include the important dependent variables that are adjusted for in the analyses.

Abstract methods and findings- the last sentence should include a limitation of your study design

Abstract conclusions- * Please address the study implications without overreaching what can be concluded from the data; the phrase "In this study, we observed ..." may be useful. * Please interpret the study based on the results presented in the abstract, emphasizing what is new without overstating your conclusions. * Please avoid vague statements such as "these results have major implications for policy/clinical care". Mention only specific implications substantiated by the results. * Please avoid assertions of primacy ("We report for the first time....")

Please remove the “research in context” section. At this stage, we ask that you include a short, non-technical Author Summary of your research to make findings accessible to a wide audience that includes both scientists and non-scientists. The Author Summary should immediately follow the Abstract in your revised manuscript. This text is subject to editorial change and should be distinct from the scientific abstract. Please see our author guidelines for more information: https://journals.plos.org/plosmedicine/s/revising-your-manuscript#loc-author-summary. 

The Data Availability Statement (DAS) requires revision. For each data source used in your study: 

Square brackets placement- please add a space between text and square brackets followed by a full stop. For example: xxxxxx [3-5]. 

Did your study have a prospective protocol or analysis plan? Please state this (either way) early in the Methods section. a) If a prospective analysis plan (from your funding proposal, IRB or other ethics committee submission, study protocol, or other planning document written before analyzing the data) was used in designing the study, please include the relevant prospectively written document with your revised manuscript as a Supporting Information file to be published alongside your study, and cite it in the Methods section. A legend for this file should be included at the end of your manuscript. b) If no such document exists, please make sure that the Methods section transparently describes when analyses were planned, and when/why any data-driven changes to analyses took place. c) In either case, changes in the analysis-- including those made in response to peer review comments-- should be identified as such in the Methods section of the paper, with rationale.

Please ensure that the study is reported according to the STROBE guideline, and include the completed STROBE checklist as Supporting Information. Please add the following statement, or similar, to the Methods: "This study is reported as per the Strengthening the Reporting of Observational Studies in Epidemiology (STROBE) guideline (S1 Checklist)." The STROBE guideline can be found here: http://www.equator-network.org/reporting-guidelines/strobe/ When completing the checklist, please use section and paragraph numbers, rather than page numbers.

For all observational studies, in the manuscript text, please indicate: (1) the specific hypotheses you intended to test, (2) the analytical methods by which you planned to test them, (3) the analyses you actually performed, and (4) when reported analyses differ from those that were planned, transparent explanations for differences that affect the reliability of the study's results. If a reported analysis was performed based on an interesting but unanticipated pattern in the data, please be clear that the analysis was data-driven.

Role of funding source should be added into the financial statement within the article metadata and removed from the main text

Please conclude the Introduction with a clear description of the study question or hypothesis.

Conclusions must be toned down since this is an observational study

Please present and organize the Discussion as follows: a short, clear summary of the article's findings; what the study adds to existing research and where and why the results may differ from previous research; strengths and limitations of the study; implications and next steps for research, clinical practice, and/or public policy; one-paragraph conclusion.

S1 Appendix table A1- is the date seen relevant? I imagine this is identifying along with the number of episodes, age and whether they were seen by CHW or at HC. Please amend this table. Manuscripts submitted to PLOS should not contain research participants personally-identifying information. In rare exceptions where this is unavoidable and a manuscript does contain PII, the authors should be willing and able to provide PLOS with confirmation of GDPR compliance upon request.

Map of Malawi- you may consider adding this into the main text as it gives an immediate visualisation of where the data were collected from

Comments from the reviewers:

Reviewer #1: This is a very important piece of work that merits consideration by the journal as it tries to assess the "… mortality outcomes of infants and children diagnosed and referred using clinical guidelines with or without pulse oximetry in Malawi". Recent advances in the wider scale up and implementation of pulseoximetry need to go hand in hand with data like the one presented in this manuscript, which are clear-cut. Pulseoximetry helps identify children at risk of dying. Authors conclude that "Pulse oximetry identified fatal pneumonia episodes at HCs in Malawi that would otherwise have been missed by WHO referral guidelines alone", which is a statement with which I do agree. I believe the journal should consider (if finally accepting the manuscript) the inclusion of a comment to go hand in hand with the manuscript. I only have a few minor comments to add:

* The word data should be used in plural

* Current global estimates for Pneumonia mortality are closer to 800,000 than to 900,000/year

* The fact that mortality as an endpoint was assessed at day 7 but also at day 30 means that many of these deaths may not (or may indeed) be related to the initial episode that took the patient to the health system (particularly for those at day 30). This should be further discussed, but it doesn't eliminate the validity of the association and therefore of the use of oxygen saturation as a risk stratification tool

* In this respect, is there any information (for example of verbal autopsy results) regarding the potential underlying causes of those deaths? It would be very helpful to be able to state that an elevated proportion of those deaths were secondary to respiratory problems

* I understand vaccination data were also collected. Was there any association between mortality and lack of adequate vaccination against Hib or pneumococcus?

* The findings that association with mortality is maintained with threshold 93% is also very important but has the risk to overload the (already fragile) system if all patients fulfilling this criterion are recommended for a transfer. Have you been able to conduct any economic modeling of costs associated with the two scenarios (transfer only if sat<90%, or transfer when <93%?). This would be super helpful, as I predict that costs to the system would be massively increased (and be unassumable). Were there other specific variables associated to mortality within the specific group with saturations between 90-93% which were also associated with mortality? Could the recommendation include a need to fulfil at least two of the risk factors, rather than one?

* Noting is mentioned on the absence/availability of emergency oxygen/systems for transfers. Recognizing this is a major deficit in the health systems of LMICS, it may be worth stating it as an important consideration for the future. There is an ethical dilemma of measuring oxygen saturation but not being able to provide life-saving oxygen, at least for the transport. This is perhaps the most neglected field in pneumonia research, how to ensure cheap and durable availability of oxygen for emergency transport

* Pulseoximter readings are very variable and often can produce false positive results of hypoxemia values which are not real. I understand that there was a specific training conduted, but perhaps it may be useful to add in the methods section how values obtained in the peripheral health system were considered "reliable" and robust (i.e whether you had to repeat more than once the reading, or you had to ensure the heart rate was also considered credible etc…). This is particularly important with one of your conclusions in relation to the association with mortality: "the failure to obtain an SpO2 measure using pulse oximeters in identifying otherwise unrecognized fatal childhood pneumonia cases accessing primary care". Could some of those cases be failures of the measurement technique? Of the devices used? It may seem very obvious, but this appears important to me.

Reviewer #2: PLOS Medicine Colbourn SpO2 Malawi review

The lowest denomination of low-cost factors to determine poor outcome is required. Oxygen saturation monitors will be increasingly available for use in developing health and an assessment of their impact is required. 

This first report is very useful for informing this field and provides structure to the reporting of future studies. 

Comments

Methods

Describe the process for patients to access and move through the HC system (i.e. HCW, HC, OP referral, Hospital in patients). 

Results

P10 L227. This manuscript would be helped by a figure or initial text in the results that provide cascade detail to the patients, matching and deaths to provide better context to what is reported.

i.e. N = 13814 pneumonia episodes, 6941 CHW + 5761 HC in which SpO2 <90% found in 86/6941 (1.2%) and 608/5761 (10.5%) respectively. There was matching of pneumonia episodes to mortality cohort in N=1112, 417/6941 CHW and 695/5761 HC. of which there were 29 deaths representing 2.6% of the matched episodes (16/417 CHW and 13/608).

P10 L252 The case for chest indrawing is well made. 

Page 11 L259

In light of the finding of a discrepancy in the SpO2 <90% and deaths in CHW cohort (missing 75% of deaths), please provide discussion on the use of an adult probe in a paediatric community setting and the potential for error from extraneous light and the difficulty in positioning. Please also consider discussion that 

Table 1 

The results identify that an SpO2 threshold of 93% would result in a 6 times higher referral rate by CHW and double referral rate by HC. Please discuss the potential impact of this on the ability to deliver safe and effective care to children with pneumonia with increased sensitivity to identify severe disease, but the potential impact of reduced sensitivity having an economic and hospital bed impact that could reduce the ability to priortise services for those with severe disease. 

Discussion

Key finding is that study identifies that clinical criteria including chest indrawing together with SpO2 <90% fail to identify 75% of deaths from pneumonia in children. It is difficult to understand that 75% of deaths occurred in children whose parents were concerned enough to attended services so early that they had no severe signs and no desaturation, but then did not re-attend when things got worse. As this represents such a large proportion of 'inaccessible pneumonia deaths' it would be useful to consider how this may be resolved in a future study, i.e red flag information. 

Please discuss that health seeking behaviours are orientated to younger children and it is older children that do not attend out patient services after referral. Whilst this is regrettable, parents so appear to understand that younger children are at greater risk. 

Overall the study is of value for providing an indication of the potential value of pulse oximetry in childhood pneumonia - but large missing data and relatively few deaths limit the strength of conclusions despite rigorous analysis. The conclusions could be more circumspect. Line 300 - This evidence 'could' support… The evidence in itself is not strong enough to support the inclusion at this time. Line 420 'should' is inappropriate based on this evidence alone. 

Reviewer #3: Alex McConnachie, Statistical Review

Colbourn et al consider the potential for the use of pulse oximetry for identifying children likely to die from pneumonia in Malawi. This review looks at the use of statistics in the paper.

Unfortunately, I have a number of concerns.

A major problem is the very poor linkage rate, which casts doubt on the validity of the results. The authors comment that there were some differences between those matched and not matched, with reference to the tables and appendix, but should perhaps expand on what these differences were in the main text.

My main concern with the analysis is the regression modelling. There were problems with convergence, and the models that it had been intended to fit (allowing for clustering, and including additional covariates and interaction terms) were not possible. There were generally few events, and the models that were fitted produced estimates with very wide confidence intervals. Some models reported in the supplement only worked using a logit link. All of these factors suggest that these models are unstable, and I would not trust these results.

Note that the description of the model is not entirely correct. On line 184, mu.i is not death for subject i, it is the probability of death; Yi is not the log risk of death, it is the outcome, death, for subject i.

Nevertheless, there may be valuable data here, but the tables as presented are quite complex, and are not easy to follow. A simpler approach might be to report more standard measures of diagnostic performance, i.e. sensitivity, specificity, PPV, NPV. That might show, in terms that are easily understood, which combinations of screening criteria (WHO, Malawi, SpO2 <90%, or <93%) worked best. I doubt that the sample size is enough to test say whether any differences are statistically significant, but might make the case for additional research.

Overall, the conclusions of the paper, that SpO2 measurements (or lack of), and chest-indrawing should be included in screening criteria, seems premature given these data.

Minor points on Table 1: p values of "0.000" should be reported as "<0.001", and there is no need to add asterisks - the actual p-values are reported.

Table 2 is confusing - some percentages are in columns, some in rows, but as stated above, if the focus were changed to measures of diagnostic performance, that might help/

Tables 3 and 5 mix up risks and odds, with the constant term referred to as baseline odds.

I note that individual consent was not obtained for this study, but individual patient data is reported in the supplement, which could potentially identify individuals.

Reviewer #4: page 6 the phrase chest in drawing might likely is the same as retractions, a term more familiar to north american. Please add

page 7 it is customary to add the name and address and even model number of a device used in research. Please add

Also, please detail why an adult universal probe was used in young children. did it work better than pediatric one or thats all there was? Thank you.

Page 10. Please define what SpO2 eligible means

Page 13. Please expand the discussion about specific limitation with SpO2 measure in your specific setting. It's what people would likely want to read in the paper.

[LINK]

---

## [Decision Letter · Decision Letter 2]

17 Jul 2020

Dear Dr. Colbourn,

Thank you very much for submitting your manuscript "Predictive value of pulse oximetry for mortality in infants and children presenting to primary care with clinical pneumonia in rural Malawi: a data linkage study" (PMEDICINE-D-20-00470R2) for consideration at PLOS Medicine. 

[LINK]

In light of these reviews, I am afraid that we will not be able to accept the manuscript for publication in the journal in its current form, but we would like to consider a revised version that addresses the reviewers' and editors' comments. Obviously we cannot make any decision about publication until we have seen the revised manuscript and your response, and we plan to seek re-review by one or more of the reviewers. 

We expect to receive your revised manuscript by Jul 27 2020 11:59PM. Please email us (plosmedicine@plos.org) if you have any questions or concerns.

We look forward to receiving your revised manuscript. 

Sincerely,

Clare Stone, PhD

Acting Chief Editor 

PLOS Medicine

plosmedicine.org

As you will see, the statistical referee raises significant issues and advises us to reject the manuscript. We have discussed this and feel that we would like to offer another revision opportunity. There is a sense that the presentation is simply trying to stretch limited data too far in producing clinical recommendations - for example, under "what do these findings mean?" all three points are "should be used" or similar, and it seems that "our findings suggest that pulse oximetry could be beneficial... and should be further investigated", say, would be more sensible. 

In addition, please do address all of the points from ref 3 (as well as from other refs). Ref 3 is our statistician and we see these points as important. I realise in some instances toning down will be a compromise. 

Comments from the reviewers:

Reviewer #1: all my comments have been addressed, and I'm pleased how authors have rewritten the manuscript. I'm also happy how they have dealt with other reviewer's comments. I still think, however, that this manuscript would benefit from an accompanying commentary.

Reviewer #2: Thank you for responding to the primary review. The responses were very helpful and provide clarity.

I have one remaining comment. Whilst I note that the authors have provided detail on the training for CHW in pulse oximetry, that '12 (75% of the 16 deaths) 'CHW may have missed danger signs, meaning further support for training...' could be inferred that CHW may somehow be considered responsible for these omissions as the children were not 'hypoxemic'. 

I would ask that the authors also acknowledge within their limitations, that the use of adult probes on pulse oximeters was a limitation and some of these children may have been hypoxemic at the time of review - we would not use adult probes where a paediatric probe is available - as is increasingly the case globally. The use of adult probes for young children have documented limitations. This potential technological failure to detect hypoxemia does not detract from their core message - but to me adds to it - as it implies that the use of age appropriate sensors may likely provide even greater sensitivity to detect children who may die from lower respiratory tract infection. 

Reviewer #3: Alex McConnachie, Statistical Review

I thank Colbourn and colleagues for their responses to my original comments. 

Seeing Table 6 (sensitivity etc.) helps me to understand the data a little better. Pulse oximetry adds sensitivity, at the expense of a loss of specificity. The results section of the paper concentrates on sensitivity, but the loss of specificity it is touched on in the discussion. No confidence intervals are provided for any of the diagnostic measures, but the authors do note that none of the differences are statistically significant.

The diagnostic odds ratios in table 6 are:

A1: 1.26, A2: 1.17, A3: 1.32

B1: 7.13, B2: 6.19, B3: 5.24

B4: 6.57, B5: 6.82, B6: 3.43

This suggests that pulse oximetry may be of no real value in the community setting, but might be in a HC setting, particularly when added to the WHO clinical criteria. It also suggests that the Malawi danger signs may perform better than the WHO danger signs on their own.

These data are limited though, because they are derived from real world data in which children are being assessed and treated. The aim of the assessments and subsequent treatment and referrals will be aiming to prevent adverse outcomes. It cannot be known how many of these children might have died in the absence of treatment. So, the actual number of true positives will be higher, and the number of false negatives lower, than suggested by simply looking at death as the outcome. Using referral and hospitalisation data as in Tables 4 and 5 does not really help, because these decisions are being made in the light of the assessments being done; in this study, there is no "gold standard" against which to assess the alternative screening criteria.

I still feel the logistic regression models are pushing the data too far. For death as the outcome, there are only 13 or 16 events being analysed, depending on the dataset, and models are being fitted (or are being attempted) with a total of 6 parameters (intercept, SpO2 (2 df), clinical danger signs, interaction (2 df)), so the events-per-variable ratio is very low. Some of the interaction terms are not estimable due to the lack of events in some subgroups. When looking at referral and hospitalisation as the outcome, there are more events, but still very few in some subgroups. The text of the paper mentions associations with low SpO2 and with clinical assessments, but these are the main effects, and each applies only in the absence of the other. The interaction terms are not taken into account, and these are generally below 1. 

I think these data are interesting, and suggest that chest in-drawing and SpO2 measurements may be of value in some settings, but it is quite messy data, which was not collected with this analysis in mind.

The linkage may be highly novel, but the success rate was very low, with many differences between those linked and not linked.

[LINK]

---

## [Editor Report · Decision Letter 3]

11 Aug 2020

Dear Dr. Colbourn,

Thank you very much for re-submitting your manuscript "Predictive value of pulse oximetry for mortality in infants and children presenting to primary care with clinical pneumonia in rural Malawi: a data linkage study" (PMEDICINE-D-20-00470R3) for review by PLOS Medicine.

I have discussed the paper with my colleagues and the academic editor. I am pleased to say that provided the remaining editorial and production issues are dealt with we are planning to accept the paper for publication in the journal.

[LINK]

We look forward to receiving the revised manuscript by Aug 18 2020 11:59PM. 

Sincerely,

Clare Stone, PhD

Managing Editor 

PLOS Medicine

plosmedicine.org

Requests from Editors:

Data – the data statement is now fine, but please remove “We can make available the matched/unmatched data actually used for the analysis (just the relevant variables, with personal identifiable information removed) and could upload that to a free to access public repository such as the one held by the corresponding authors university UCL. Please let us know if this is acceptable.”

Comments from Reviewers:

[LINK]

---

## [Editor Report · Decision Letter 4]

11 Sep 2020

Dear Dr. Colbourn, 

On behalf of my colleagues and the academic editor, Dr. Quique Bassat, I am delighted to inform you that your manuscript entitled "Predictive value of pulse oximetry for mortality in infants and children presenting to primary care with clinical pneumonia in rural Malawi: a data linkage study" (PMEDICINE-D-20-00470R4) has been accepted for publication in PLOS Medicine. 

PRODUCTION PROCESS

PRESS

PROFILE INFORMATION

Thank you again for submitting the manuscript to PLOS Medicine. We look forward to publishing it. 

Best wishes, 

Clare Stone, PhD

Managing Editor 

PLOS Medicine

plosmedicine.org